# *IS481EU* Shows a New Connection between Eukaryotic and Prokaryotic DNA Transposons

**DOI:** 10.3390/biology12030365

**Published:** 2023-02-25

**Authors:** Kenji K. Kojima, Weidong Bao

**Affiliations:** Genetic Information Research Institute, Cupertino, CA 95014, USA

**Keywords:** DNA transposon, insertion sequence, DDD/E transposase, IS*481*, *IS481EU*, target site duplications (TSDs), parabasalids

## Abstract

**Simple Summary:**

Transposable elements (TEs) are genetic parasites that mobilize themselves from one locus to another in the host genome. DDD/E transposase gene is the most abundant gene in nature and its protein product catalyzes DNA transposition. The IS*481* family is a group of prokaryotic TEs that encode a DDD/E transposase. Here, we report a group of eukaryotic TEs that shows close affinity to some of the prokaryotic IS*481* family members, and designate them *IS481EU*. *IS481EU* was found in palabasalids including *Trichomonas vaginalis*. Although most TEs with DDD/E transposase generate direct repeats of fixed or similar lengths (target site duplications, TSDs) at both ends of insertion, *IS481EU* generates TSDs of discrete lengths (~4 bps, ~15 bps, or ~25 bps). The unique characteristics of *IS481EU* in protein sequences and the distribution of TSD lengths support its placement as a new superfamily of eukaryotic DNA transposons. The transmission of prokaryotic IS*481* to a eukaryotic lineage likely resulted in the birth of *IS481EU*, and may have contributed to lineage-specific evolutionary trajectories.

**Abstract:**

DDD/E transposase gene is the most abundant gene in nature and many DNA transposons in all three domains of life use it for their transposition. A substantial number of eukaryotic DNA transposons show similarity to prokaryotic insertion sequences (ISs). The presence of IS*481*-like DNA transposons was indicated in the genome of *Trichomonas vaginalis*. Here, we surveyed IS*481*-like eukaryotic sequences using a bioinformatics approach and report a group of eukaryotic IS*481*-like DNA transposons, designated *IS481EU*, from parabasalids including *T. vaginalis*. The lengths of target site duplications (TSDs) of *IS481EU* are around 4 bps, around 15 bps, or around 25 bps, and strikingly, these discrete lengths of TSDs can be observed even in a single *IS481EU* family. Phylogenetic analysis indicated the close relationships of *IS481EU* with some of the prokaryotic IS*481* family members. *IS481EU* was not well separated from *IS3EU*/*GingerRoot* in the phylogenetic analysis, but was distinct from other eukaryotic DNA transposons including *Ginger1* and *Ginger2*. The unique characteristics of *IS481EU* in protein sequences and the distribution of TSD lengths support its placement as a new superfamily of eukaryotic DNA transposons.

## 1. Introduction

Transposable elements (TEs), transposons or mobile DNA, include a wide variety of DNA segments that can, in a process called transposition, move or duplicate themselves from one location to another [1]. TEs are traditionally grouped into two groups: retrotransposons and DNA transposons [2]. DDD/E transposase/integrase is the most abundant gene in nature [3], and is the most common transposase in both eukaryotic and prokaryotic DNA transposons [4]. DDD/E transposases constitute the RNase H fold [5] and contain three catalytic residues (DDD or DDE) for DNA strand transfer [4]. Upon integration, a short DNA segment is duplicated at both sides of DNA transposons and they are called target site duplications (TSDs). Integrases encoded by long terminal repeat (LTR) retrotransposons and retroviruses are also a type of DDD/E transposases [6]. Recombination-activating gene 1 (RAG1), the catalytic component for the V(D)J recombination system of jawed vertebrates, is a domesticated DDD/E transposase derived from a DNA transposon [7,8].

An insertion sequence (IS) was originally defined as a short DNA segment encoding only the enzymes necessary for its transposition [9,10,11]. Around 30 IS “families” are designated at ISFinder (https://www-is.biotoul.fr/, accessed on 23 May 2022), and the majority of them encode a DDD/E transposase or its variety. Sharing additional conserved residues apart from the DDE motif, some IS families are more closely related to each other than to others. The transposases of the IS*3* family, the IS*481* family, and the IS*1202* group show a certain level of similarity to one another. Currently the IS*1202* group is classified inside the IS*NCY* (IS not classified yet) family. Known eukaryotic DNA transposons are classified into 23 “superfamilies” in Repbase (https://www.girinst.org/repbase/, accessed on 24 May 2022) [12,13]. Among them, 21 superfamilies encode a DDD/E transposase. Like prokaryotic IS families, different eukaryotic transposon superfamilies sometimes share conserved motifs. For example, *Dada*, *hAT*, *Kolobok*, *MuDR*, and *P* share the signature motif “C/DxxH” between the second D and the last E conserved residues [14,15].

Despite the ancient splitting of eukaryotes and prokaryotes, a notable level of similarity is observed between a number of eukaryotic DDD/E transposases and their prokaryotic counterparts. IS*630* in prokaryotes and *Mariner*, *Tc1*, *pogo*, and many related DNA transposons in eukaryotes have related DDD/E transposases, and are altogether called I*Tm* [16]. Besides the sequence similarity of transposases, they generate 2 bp TSDs of TA in general. Multiple horizontal transfers from prokaryotes to eukaryotes have been suggested in this superfamily [17]. Prokaryotic IS*1380* and eukaryotic *piggyBac* are distantly related, and they share 4 bp TSDs and the 5′-CC..GG-3′ termini [18,19]. Eukaryotic *Merlin* DNA transposons show a clear relationship with the prokaryotic IS*1016* group inside of the IS*1595* family [20]. Both generate 8 bp TSDs. Eukaryotic *Zator* shows a similarity to the prokaryotic IS*Azo13* family and both of them generate 3 bp TSDs [21,22]. The relationship between prokaryotic IS*256* and eukaryotic *MuDR* was reported and they share a signature of conserved residues in addition to the catalytic core with other eukaryotic DNA transposons [14,15,23,24]. They generate 4 bp to 10 bp TSDs. Eukaryotic *Harbinger* and *ISL2EU* superfamilies, as well as their relatives, called *Spy*, *Nuwa*, *Pangu, and ISL2PR* show a similarity to prokaryotic IS*L2* and IS*5* and together constitute *PHIS* [25,26,27]. They generate no TSDs or 1 bp to 4 bp TSDs. The horizontal transfer of prokaryotic IS*5* from bacteria to the genome of bdelloid rotifer was also reported [26]. The transposase similarity and the similar lengths of TSDs suggest a shared mechanism of transposition, even between eukaryotic and prokaryotic TEs.

In our previous article, we reported three proteins that show similarity to the prokaryotic IS*481* transposases from the genome of *Trichomonas vaginalis*, which was designated as *IS481EU* [28]. The presence of DNA transposons encoding an IS*481*-like transposase was also indicated and designated as *Banshee* [29]. However, these sequences have not been analyzed further and few characteristics of these putative transposons have been reported to date. Here, we report a group of eukaryotic DNA transposons showing similarity to the prokaryotic IS*481* family from the multiple genomes of palabasalids, including *T. vaginalis*. One of the distinctive features of these *IS481EU* DNA transposons is the generation of three distinct lengths of TSDs: around 4 bp, around 15 bp or around 25 bp TSDs. The phylogenetic analysis suggested the close affinity of *IS481EU* with a sublineage of the IS*481* family. *IS481EU* was not separated from the *IS3EU/GingerRoot* superfamily of eukaryotic DNA transposons in our phylogenetic analysis, but the different features of proteins and TSDs support its position as a new superfamily of eukaryotic DNA transposons.

## 2. Materials and Methods

### 2.1. Characterization of IS481EU Families

Censor [30] searches were performed against the genome of *T. vaginalis* with the protein sequences reported in [28]. Censor hits were extracted and clustered with BLASTCLUST 2.2.25 in the NCBI BLAST package with the thresholds at 75% length coverage and 75% sequence identity. The consensus sequence for each cluster was generated with the 50% majority rule applied and the help of homemade scripts. Censor searches were performed with the consensus sequence of each cluster against the genome. Up to 10 Censor hits were extracted with 5000 bp flanking sequences at both sides. Consensus sequences were regenerated to be elongated to find both termini. The termini were determined based on the terminal TG..CA signatures and the presence of TSDs.

Censor searches were performed with the protein sequences of *IS481EU* from *T. vaginalis* against the genomes of palabasalids (Table 1). Genome sequences were downloaded from NCBI Assembly (https://www.ncbi.nlm.nih.gov/assembly, accessed on 18 May 2022). The characterization of complete *IS481EU* sequences was conducted similarly to the cases of *IS481EU* from *T. vaginalis*. All characterized consensus sequences are available as Appendix A and have also been submitted to Repbase (https://www.girinst.org/repbase/, accessed on 24 May 2022) [12].

### 2.2. Protein Dataset and Phylogenetic Analysis

All protein sequences of *IS481EU* families from *T. vaginalis* and *Histomonas meleagridis* were used for the phylogenetic analysis. All protein sequences of the IS*481* family and the IS*1202* group were extracted from ISFinder (https://www-is.biotoul.fr/, accessed on 24 May 2022). Representatives of the IS*3* family were chosen based on [11], and were extracted from ISFinder (https://www-is.biotoul.fr/, accessed on 23 May 2022) [9]. The protein sequences of all autonomous *IS3EU* (17 entries), *Ginger1* (23 entries), and *Ginger2* (30 entries) were extracted from Repbase (https://www.girinst.org/repbase/, accessed on 24 May 2022) [12]. The protein sequences of *GingerRoot* were extracted from the dataset reported in the original article [31]. Representatives of LTR retrotransposons and endogenous retroviruses were obtained from Repbase (https://www.girinst.org/repbase/, accessed on 24 May 2022) [12]. The sequences of HIV-1 and HTLV-1 were extracted from the dataset available at GyDB (https://gydb.org/, accessed on 24 May 2022) [32]. They were aligned with the help of MAFFT v.7.407 [33] with the linsi option or Clustal Omega [34] in the Seaview package. The conserved transposase domains were extracted and realigned with Probcons version 1.12 [35]. Protein sequences with a large deletion were removed.

A maximum likelihood tree was generated on the PhyML 3.0 server (http://www.atgc-montpellier.fr/phyml/, accessed on 27 May 2022) [36] with 100 bootstrapping supports and 2 datasets. One dataset was composed of the *IS481EU* and IS*481* proteins. The substitution model Q.pfam +G+I was chosen based on the Bayesian information criterion (BIC). The other dataset was composed of the representatives of *IS481EU*, IS*481*, IS*1202*, IS*3*, *IS3EU*, *Ginger1*, *Ginger2*, *GingerRoot*, LTR retrotransposons, and retroviruses. The substitution model Q.pfam +G+I+F was chosen based on the BIC. The phylogenetic tree was rooted at the midpoint and visualized with FigTree v.1.4.3 (http://tree.bio.ed.ac.uk/software/figtree/, accessed on 27 May 2022).

## 3. Results

### 3.1. Eukaryotic TEs Related to the Prokaryotic IS481 Family

In the survey of TEs encoding a protein similar to the reported IS481EU transposases in the genome of a protist T. vaginalis, we were able to characterize 10 autonomous IS481EU families and 7 non-autonomous families (Table 1 and Appendix A). The letter N in the family name indicates a non-autonomous family. While we could characterize the autonomous counterparts of four non-autonomous families (for example, IS481EU-4_TV for IS481EU-4N1_TV), we could not find the autonomous counterparts of the remaining three non-autonomous families (IS481EU-N1_TV to IS481EU-N3_TV). These IS481EU transposons from T. vaginalis show the terminal signature TNT..AYA (Figure 1). All IS481EU families have terminal inverted repeats (TIRs), mostly shorter than 50 bps. All autonomous families encode a single protein containing a DDD/E transposase. The most abundant IS481EU family is IS481EU-7_TV and it has 376 copies which retain both termini. One of the interesting characteristics of IS481EU is the distinct lengths of TSDs. Several families of IS481EU, such as IS481EU-1_TV and IS481EU-2_TV generate 15 bp or 16 bp TSDs, while other families such as IS481EU-3_TV and IS481EU-5_TV generate 4 bp TSDs. IS481EU-6_TV and IS481EU-8_TV show both 4 bp TSDs and around 15 bp TSDs. The majority of IS481EU-7_TV copies are flanked with 4 bp TSDs; however, two copies are flanked with 15 bp TSDs, and one copy with 24 bp TSDs. Non-autonomous IS481EU families show the same features in the lengths of TSDs. Banshee was reported to generate 4 bp or 15 bp TSDs and starts with TGT [29], so Banshee is likely a family of IS481EU.

Homology search at the NCBI BLAST website (https://blast.ncbi.nlm.nih.gov/Blast.cgi, accessed on 18 May 2022) with the transposases of *IS481EU* families revealed the presence of similar transposases in the genomes of *Trichomonas tenax*, *Trichomonas gallinae*, *Tritrichomonas foetus*, and *Histomonas meleagridis* (Table 1 and Appendix A). Although most of these transposases were found as a single copy in the respective genome, we were able to reconstruct five complete DNA transposon consensus sequences from the genome of *H. meleagridis. Trichomonas* and *Histomonas* belong to the different orders inside palabasalids. We investigated the TSDs of *IS481EU* from *H. meleagridis* to check whether they share the same features as *IS481EU* from *T. vaginalis* (Figure 2). All *IS481EU* families from *H. meleagridis* show TGT…ACA termini, and many of their copies generate 15 bp or 16 bp TSDs upon integration. *IS481EU-5_HisMel* has a non-autonomous family (*IS481EU-5N1_HisMel*) and one of its copies is flanked by degenerated 24/25 bp TSDs.

### 3.2. The Phylogenetic Relationship between IS481EU and IS481

We performed a phylogenetic analysis of *IS481EU* with all prokaryotic IS*481* elements available on ISFinder (Figure 3). The IS*481* family is distributed among both bacteria and archaea. The IS*481* family generates 2 to 15 bp TSDs, but the majority of IS*481*s generate 6 bp TSDs (ISFinder, https://www-is.biotoul.fr/, accessed on 23 May 2022). Three large groups composed of prokaryotic IS*481* elements generating 6 bp TSDs were statistically supported (Bootstrap values: 89%, 98%, and 83%). The other two prokaryotic lineages were supported statistically and were composed of ISs with different lengths of TSDs.

Families of *IS481EU* clustered together with ample statistical support (77% bootstrap value). Based on the phylogeny, the ancestral *IS481EU* family seems to have generated 15 bp TSDs. Two sublineages of *IS481EU* show 4 bp TSDs. One is the sublineage composed of three families: *IS481EU-3_TV*, *IS481EU-4_TV*, and *IS481EU-5_TV*. The other is composed of *IS481EU-9_TV* and *IS481EU-10_TV*. Its sister family, *IS481EU-7_TV*, is an exceptional family that generates 4 bp, 15 bp, and 24 bp TSDs. Another family, *IS481EU-6_TV* generates both 4 bp and 15 bp TSDs, but 4 bp TSDs are dominant. The phylogeny indicates that the change in TSD length has occurred multiple times in the evolution of *IS481EU*. The mixed phylogenetic positions of *IS481EU from T. vaginalis and H. meleagridis* suggests the presence of multiple lineages of *IS481EU* in the evolution of parabasalids, although the current data cannot exclude the possibility of the horizontal transfer of *IS481EU* between parabasalids.

Although the statistical support is not high enough, two lineages of prokaryotic IS*481* elements are positioned close to the *IS481EU* lineage. IS*Mac4* and IS*A0963-1* are from archaea and generate 15 bp TSDs. IS*Tni2* and IS*Chy3* generate 6 bp TSDs and are found in bacteria. Unfortunately, the current data cannot clarify the origin of *IS481EU*.

### 3.3. The Origins of IS3EU/GingerRoot and IS481EU in Eukaryotes

The BLASTP search on ISFinder (https://www-is.biotoul.fr/, accessed on 23 May 2022) with the transposase of IS481EU-1_TV found its similarity to many IS3 family members, as well as IS481 family members. IS481 was originally considered a member of IS3 [10] and, thus, the sequence similarity between these two families is expected. Besides these two groups, three ISs (ISArsp14, ISBli29, and ISHahy13) were hit. All of them belong to the ISNCY (IS not classified yet) family. ISHahy13 is a compact IS and belongs to the IS1202 group inside ISNCY. IS1202 is reported to be distantly related to IS481 [10]. ISArsp14 and ISBli29 are large ISs with passenger genes. These three ISs terminate with 5′-TG and CA-3′, as with IS3 and IS481. We included the three groups of ISs (IS481, IS3, and IS1202) for the phylogenetic analysis.

Our previous analysis supported the independent origin of IS481EU from *Ginger1*, *Ginger2*, or Polinton [28]. Since the report, two groups of eukaryotic TEs have been reported: IS3EU and GingerRoot. As a designated superfamily name, IS3EU have been listed in Repbase (https://www.girinst.org/repbase/, accessed on 24 May 2022) since 2013. Now, there are 45 IS3EU families stored in Repbase as of 24 May 2022, 17 of which are autonomous [12]. The complete IS3EU families are always associated with TA..TA termini. TSDs are usually 6 bp long. GingerRoot was characterized as a new group of TEs showing similarity to *Ginger1* and *Gigner2* from the clubmoss Selaginella lepidophylla, and related TEs are supposed to be present in animals too [31]. GingerRoot families generate 6 bp TSDs, and their termini are mostly TA..TA and occasionally TG..CA. Since there is a certain level of sequence similarity between IS3 and IS481, the relationship between IS481EU, IS3EU, and GingerRoot was investigated (Figure 4).

*IS481EU*, *IS3EU*, *GingerRoot*, *Polinton*, IS*3*, IS*481*, and IS*1202* families do not have an HHCC motif upstream of the transposase core unlike the integrase of HIV-1 [28]. The N-terminal domain of integrase, which includes an HHCC motif, coordinates the binding of a zinc ion to form a structure similar to those of helix-turn-helix DNA-binding domains, and stimulates catalytic activity [37]. The first catalytic D is located 100 to 300 residues downstream from the N-terminus in *IS481EU*, *IS3EU*, IS*3*, IS*481*, and *IS1202*, and it is possible that the N-terminal region of these transposases is functionally equivalent to the N-terminal domain of integrase. The N-terminal regions of *IS481EU* proteins show a weak similarity to helix-turn-helix DNA-binding proteins such as TetR transcriptional regulators, according to the results of HHpred, protein homology detection, and structure prediction software [38]. Only the transposase core regions were able to be aligned reliably (Appendix A).

Due to the low sequence conservation and the short length of transposases, only a few lineages were statistically supported. First, *IS3EU* and *GingerRoot* belong to the same lineage with strong statistical support (Bootstrap value: 99%), and thus this lineage is designated as *IS3EU*/*GingerRoot* hereafter. Second, the monophyly of the entire IS*1202* family is supported (Bootstrap value: 96%). Third, the IS*3* family is separated into three lineages, and two of them are well supported. The monophyly of the IS*2* subgroup is highly supported (Bootstrap value: 100%), and the monophyly of the IS*407* subgroup is supported too (Bootstrap value: 84%). The clustering of IS*3*, IS*51*, and IS*150* subgroups is moderately supported (53%).

As we have no obvious outgroup, the root of this phylogenetic tree cannot be determined. The presence of the N-terminal HHCC domain of integrase in LTR retrotransposons, retroviruses, and *Ginger1* [28], suggests the common origin of these transposases and integrases, and so, the basal positions of some LTR retrotransposons could be artifacts. Although there is no statistical support, the distinct positions of two lineages of eukaryotic TEs indicate at least two independent origins of eukaryotic TEs. One is composed of *IS481EU* and *IS3EU/GingerRoot*, and the other is of *Ginger1*, *Ginger2*, *Polinton*, LTR retrotransposons, and retroviruses. *IS3EU/GingerRoot* and *IS481EU* are not separated. The close affinity of IS*Tni2*, IS*A0963_1*, and IS*Mac4* with *IS481EU* is again seen in this phylogenetic analysis, although there is no significant statistical support.

## 4. Discussion

Here, we report *IS481EU*, a new group of eukaryotic DNA transposons encoding a DDD/E transposase. The origin of *IS481EU* could not be distinguished from that of *IS3EU/GingerRoot* but is likely distinct from the origin of *Ginger1*, *Ginger2*, *Polinton*, LTR retrotransposons, and retroviruses. Although there is still a possibility that *IS481EU* and *IS3EU/GingerRoot* share a eukaryotic ancestral DNA transposon, based on the distinct characteristics of these two groups, we propose *IS481EU* as a new superfamily of eukaryotic DNA transposons. The close relationship of *IS481EU* to some of prokaryotic IS*481* members suggests the horizontal transfer of an IS*481* family member to a certain eukaryote contributed to the birth of *IS481EU*. This is the third superfamily of eukaryotic transposons designated based on the similarity to the prokaryotic ISs, following *ISL2EU* and *IS3EU* [12]. We can expect that as more genomes are sequenced, there will be a greater chance that new *ISxEU*s would be identified. Thus, it is tempting to introduce a new term MILET (Minor IS-like Eukaryotic Transposon Superfamilies) to refer to these “minor” superfamilies which may have spread at a later evolutionary time or only achieved limited success in their transition from prokaryotes to eukaryotes.

The relationship between *IS481EU* and *Banshee* should be mentioned. Although there is no sequence information publicly available for *Banshee*, it is reported to generate 4 or 15 bp TSDs and started with 5′-TGT [29]. *Banshee* was found only in *T. vaginalis* [10]. We could not find any other TE families related to IS*481* other than *IS481EU* from *T. vaginalis*. These results strongly suggest that the reported *Banshee* is a member of *IS481EU*.

TEs belonging to the same superfamily tend to generate TSDs of similar lengths, which could be an indicator of the associated superfamily [13,39]. In the case of IS*3* family, there are three types of TSD length and they correlate with the subgroups inside the IS*3* family [9]. The IS*407* subgroup is associated with 4 bp TSDs, and the *IS2* subgroup is associated with 5 bp TSDs. Other IS*3* family members are flanked with 3 bp TSDs. The characteristic TSD lengths are 4 bp for *Ginger1*, 4 or 5 bp for *Ginger2*, 6 bp for *Polinton*, and 3–6 bp for LTR retrotransposons and retroviruses [28,39,40].

Retroviral integrase recognizes scissile phosphodiester bonds in target DNA which are separated by 4 to 6 bp for strand transfer [41]. The target DNA is significantly kinked to optimally position the catalytic site for the pairwise strand transfer events in the complex. Severe target DNA bending is also observed in transpososomes of various superfamilies of DNA transposons including bacterial Mu, IS*21*, and eukaryotic *Mos1* [42,43,44]. The single-stranded gaps are repaired by cellular enzymes and result in TSDs. Thus, different degrees of target DNA bending along with the unique distance of two catalytic sites would lead to the characteristic lengths of TSDs. Conserved lengths of TSDs in each superfamily of DNA transposons and LTR retrotransposons suggest the conservative nature of transposase–DNA interaction.

Contrary to the relatively conserved TSD lengths among the IS*3* family and the superfamilies of eukaryotic DNA transposons related to them, the IS*481* and IS*1202* families show discrete lengths of TSDs among members. The majority of IS*481* family members generate 6 bp TSDs, but not a few IS*481* family members generate ~15 bp TSDs (ISFinder: https://www-is.biotoul.fr/, accessed on 23 May 2022). The IS*1202* group members show three separate ranges of the length of TSDs: 5 or 6 bp, 15 to 17 bp, and 24 to 29 bp (ISFinder: https://www-is.biotoul.fr/, accessed on 23 May 2022). The IS*1202* group is located inside the IS*481* family in the phylogeny (Figure 4). The three separate length ranges of TSDs for the IS1202 group is indicative of the conserved mechanisms to generates TSDs. Even in this circumstance, *IS481EU* is still unusual. One characteristic of *IS481EU* is the two or more discrete lengths of TSDs, which can be observed even among the copies of the same family. The extreme case is observed in *IS481EU-7_TV*. Among the 11 full-length copies, 7 were flanked with 4 bp TSDs, 1 with 5 bp TSDs, 2 with 15 bp TSDs, and 1 with 24 bp TSDs (Figure 1). We can speculate that the three discrete lengths of TSDs seen for *IS481EU*, IS*481*, and IS*1202* are generated through similar mechanisms.

How are such largely discrete lengths of TSDs generated? The situation is different from the two different lengths of TSDs seen among the *Academ* superfamily [45]. In the *Academ* superfamily, *AcademH* generates 9 or 10 bp TSDs, while *AcademX* generates 3 or 4 bp TSDs. The difference in the TSD lengths between these two *Academ* lineages seems related to the different enzymes, helicase and nuclease, encoded by them. Here, as seen in the case of *IS481EU-7_TV*, the enzymes functioning during transposition are the same that generate two or more different lengths of TSDs. Notably, the three distinct ranges of TSD lengths observed in the groups of IS*1202*, IS*481*, and *IS481EU* are close to 10 bps apart, a measurement that equals one full turn of the B-DNA double helix.

The transposases of *IS481EU* show clear similarity to retroviral integrases (Appendix A), the structure and function of which are well-studied [41]. Among the three discrete ranges of TSD lengths seen among *IS481EU*, IS*481*, and IS*1202*, it is likely that the shortest 4 to 6 bp TSDs would correspond to the TSDs of retroviruses, which are 4 to 6 bps in length. Retroviral integrase works as a tetramer, with a dimer-of-dimers architecture. The inner subunits of each dimer are responsible for interaction with DNA, while the outer subunits attach to the inner subunits but not to the target DNA. Thus, the length of TSDs is dictated by the distance between the two catalytic cores of inner subunits and the bendability of target DNA. The 4 bp TSDs observed upon *IS481EU* integration are likely to be generated similarly to the cases of retroviral integrases, by two inner subunits of *IS481EU* transposase tetramer.

We have currently no clue for how the longer TSDs are generated upon transposition of *IS481EU*. It can be speculated that the two ranges of longer TSDs (~16 bp and ~25 bp) are generated when the target DNA is cleaved by one inner subunit and one outer subunit, or by two outer subunits, with different DNA-binding surfaces. Another possibility is that *IS481EU* has different multimer conformations and different conformations result in different lengths of TSDs. Alternatively, DNA might form a loop between two binding sites. Structural and biochemical analysis is necessary to clarify the mechanisms generating different lengths of TSDs.

## 5. Conclusions

*IS481EU* is characterized from the unicellular eukaryotic lineage parabasalids, and here is proposed as a new superfamily of eukaryotic DNA transposons. It shows a new link between prokaryotic and eukaryotic TEs via the sequence similarity in their termini and transposases, as well as the TSD length variations.

## Figures and Tables

**Figure 1 biology-12-00365-f001:**
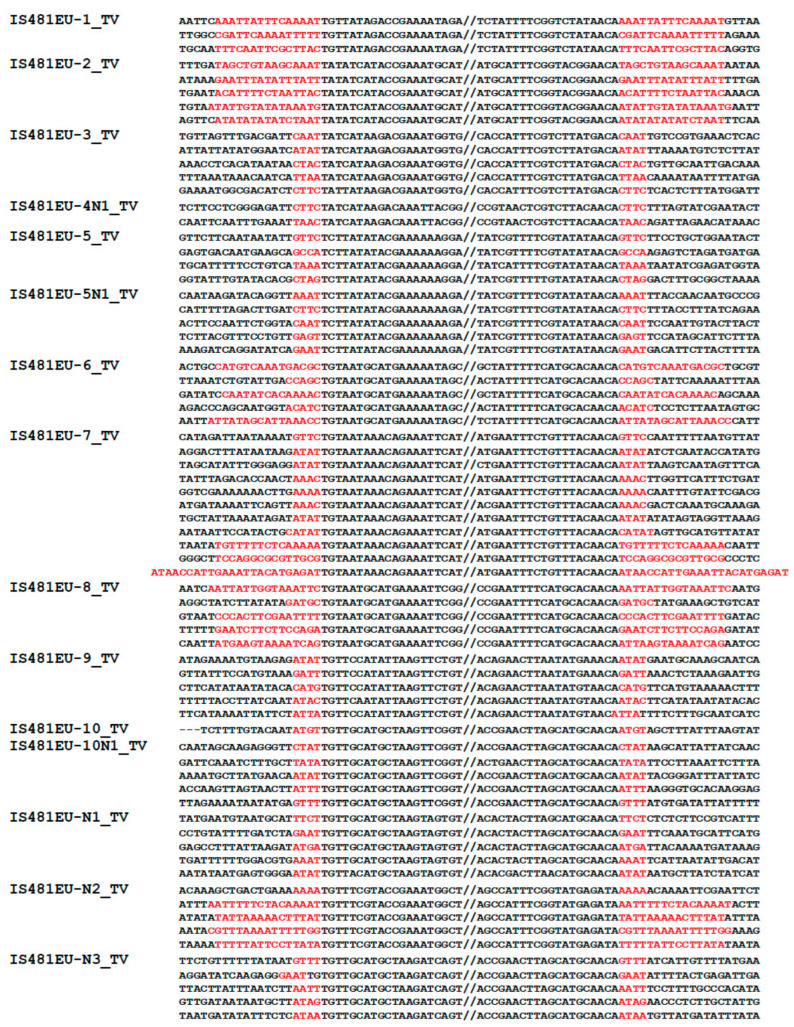
Termini and boundaries of *IS481EU* families from *T. vaginalis*. If there are fewer than five copies with both termini, all copies with TSDs are shown with flanking sequences. If there are more than five copies with both termini, only the 5 representative copies with TSDs are shown, except for *IS481EU-7_TV*, for which 10 representative copies are shown. TSDs are colored in red.

**Figure 2 biology-12-00365-f002:**
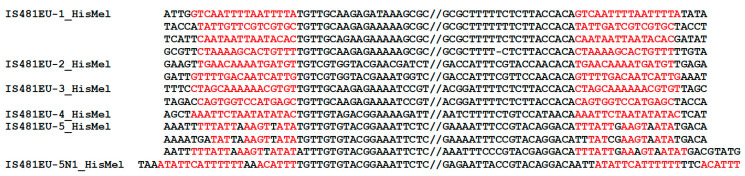
Termini and boundaries of *IS481EU* families from *H. meleagridis*. All copies with TSDs are shown with flanking sequences. TSDs are colored in red.

**Figure 3 biology-12-00365-f003:**
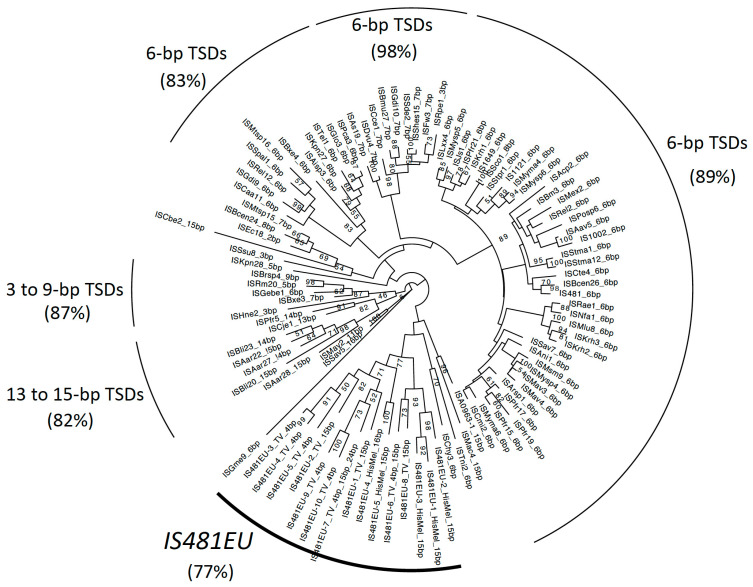
Phylogenetic tree based on the transposase domains of *IS481EU* and IS*481* elements. The length of TSDs is shown at each node after the element name. Lineages with ample bootstrap supports (over 70%) are indicated.

**Figure 4 biology-12-00365-f004:**
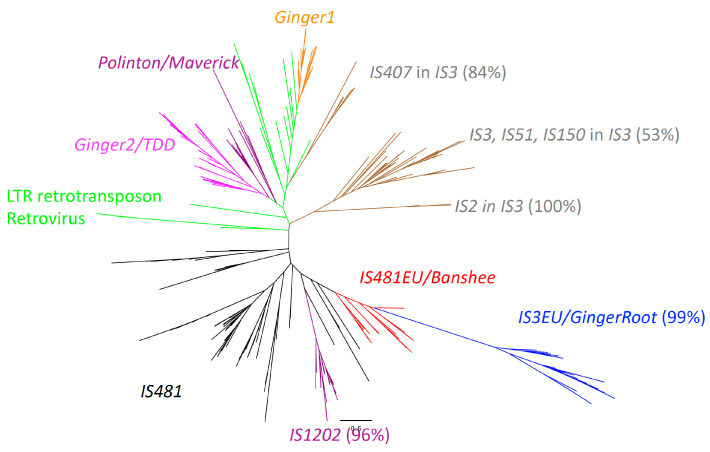
Phylogenetic tree based on the transposase domains of eukaryotic *IS481EU*, *IS3EU*/*GingerRoot, *Ginger1*, *Ginger2*/*TDD*, Polinton/Maverick* superfamilies, and prokaryotic IS*481,* IS*3,* IS*1202* families, and the integrases of LTR retrotransposons and retroviruses. Lineages with ample bootstrap supports (over 50%) are indicated. IS*2,* IS*3,* IS*51,* IS*150*, and IS*407* are subfamilies inside the *IS3* family. The colors of lineages are as follows: orange, *Ginger1*; magenta, *Ginger2*/*TDD*; purple, *Polinton*/*Maverick*; red, *IS481EU*/*Banshee*; blue, *IS3EU*/*GingerRoot*; black, IS*481*, grey, IS*3*; purple, IS*1202*; green, LTR retrotransposons and retroviruses.

**Table 1 biology-12-00365-t001:** The distribution of *IS481EU* in parabasalids.

Phylum	Order	Family	Species	Genome Assembly	*IS481EU* Family
Parabasalia	Trichomonadida	Trichomonadidae	*Trichomonas vaginalis*	ASM289133v1	*IS481EU-1_TV* to *10_TV, IS481EU-N1_TV* to *N3_TV, 4N1_TV, 5N1_TV, 8N1_TV*, and *10N1_TV.*
			*Trichomonas tenax*	PRJEB22701	*IS481EU-1_TrTenax*
			*Trichomonas gallinae*	MiGF1c1.0	*IS481EU-1_TrGallinae* to *IS481EU-5_TrGallinae*
	Tritrichomonadida	Tritrichomonadidae	*Tritrichomonas foetus*	TF_PacBio	*IS481EU-1_TrFoetus*
		Dientamoebidae	*Histomonas meleagridis*	ASM2018611v1	*IS481EU-1_HisMel* to *5_HisMel*

## Data Availability

The data presented in this study are available as Appendix A.

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
