# Peer review of "IS481EU Shows a New Connection between Eukaryotic and Prokaryotic DNA Transposons"

_biology, 2023, doi:10.3390/biology12030365_

Round 1

Reviewer 1 Report

Line 146

In a survey of “T. vaginalis, we were able to characterize 10 autonomous 147 IS481EU families and 7 non-autonomous families (Table 1)”. These are listed as “IS481EU-1_TV to 10_TV, IS481EUN1_TV to N3_TV” in table 1. Is the N in the latter names supposed to refer to non-autonomous? If so make this explicit. There seems to be a total of 13 families in Table 1 for T. vaginalis and 15 in Figure 1.

Line 172 – “All IS481EU families 172 from H. meleagridis show TGT…ACA termini, and generate 15-bp or 16-bp TSDs upon 173 integration.” Can the authors explain why some of the sequences in Figure 2 do not have TSDs highlighted?

Paragraph 236-241. The function of the HHCC motif in integrases should be explained here. The authors say “ it is possible that the N-terminal region of these 239 transposases is functionally equivalent to the N-terminal HHCC domain (rather than motif?) of integrase[s]”. Is there any indication, for example from structure prediction or searches for other motifs, what the structure and function of the N-terminal regions may be?

Line 283-289. What are the typical TSDs and termini of IS3EU and GingerRoot? This would need to be made clear, so that the reader can judge if IS481 is indeed a separate family from IS3EU and GingerRoot

Line 335-340 Another possibility is that the transpososome loops the target DNA so that it can insert into positions separated by 15/6-bp or 25-bp in the target DNA.

Line 221 - Now, there are 45 IS3EU families 221 stored in Repbase as of May 23, 2022, 17 of which are autonomous”. Could the authors provide a more up-to-date figure?

Minor points:

Line 65 - Their instead of theirs

Line 152

IS481EU-2_TV generate 15-bp TSDs (but there is also a 16-bp TSD highlighted). Can the authors comment on this?

Line 159 – “fewer” rather than “less”

 Line 228-233 Figure 4 – colour the name labels with the colour allocated in the legend.

Author Response

We would like to thank all reviewers and the editor for the thorough review of our manuscript. We have read the comments carefully and following these suggestions, we have revised the manuscript. 

Reviewer 1

Line 146

In a survey of “T. vaginalis, we were able to characterize 10 autonomous 147 IS481EU families and 7 non-autonomous families (Table 1)”. These are listed as “IS481EU-1_TV to 10_TV, IS481EUN1_TV to N3_TV” in table 1. Is the N in the latter names supposed to refer to non-autonomous? If so make this explicit. There seems to be a total of 13 families in Table 1 for T. vaginalis and 15 in Figure 1.

Response:

We are sorry for this error. We added 4 remaining non-autonomous families (IS481EU-4N1_TV, 5N1_TV, 8N1_TV and 10N1_TV) on Table 1. We added the description of non-autonomous families and their relationships to autonomous families.

Line 172 – “All IS481EU families 172 from H. meleagridis show TGT…ACA termini, and generate 15-bp or 16-bp TSDs upon 173 integration.” Can the authors explain why some of the sequences in Figure 2 do not have TSDs highlighted?

Response:

We corrected the statement. We could not find TSDs from all IS481EU insertions. We also updated figure 2 to show only IS481EU insertions with clear TSDs.

Paragraph 236-241. The function of the HHCC motif in integrases should be explained here. The authors say “ it is possible that the N-terminal region of these 239 transposases is functionally equivalent to the N-terminal HHCC domain (rather than motif?) of integrase[s]”. Is there any indication, for example from structure prediction or searches for other motifs, what the structure and function of the N-terminal regions may be?

Response:

We added some descriptions of the functions of N-terminal domain of integrase (We corrected “HHCC domain” with “N-terminal domain” or “HHCC motif”). Our HHpred analysis indicates that the N-terminal region of IS481EU proteins shows a helix-turn-helix structure resembling some HTH proteins, such as TetR transcription factors.

Line 283-289. What are the typical TSDs and termini of IS3EU and GingerRoot? This would need to be made clear so that the reader can judge if IS481 is indeed a separate family from IS3EU and GingerRoot

Response:

Thank you for the suggestions. We added the information on TSD lengths and termini of IS3EU and GingerRoot.

Line 335-340 Another possibility is that the transpososome loops the target DNA so that it can insert into positions separated by 15/6-bp or 25-bp in the target DNA.

Response:

Thank you for the suggestion. We added some text for the mechanism to generate different lengths of TSDs. “Or DNA might form a loop between two binding sites. Biochemical analysis is necessary to clarify the mechanisms generating different lengths of TSDs.”

Line 221 - Now, there are 45 IS3EU families 221 stored in Repbase as of May 23, 2022, 17 of which are autonomous”. Could the authors provide a more up-to-date figure?

Response:

There is no new addition of IS3EU to Repbase in the year 2022. We internally have some IS3EU sequences, but unfortunately, they need an additional round of curation in order to be published.

Minor points:

Line 65 - Their instead of theirs

Response:

Corrected.

Line 152

IS481EU-2_TV generates 15-bp TSDs (but there is also a 16-bp TSD highlighted). Can the authors comment on this?

Response:

We rephrased the text by “15-bp or 16-bp TSDs”. It is common that slightly different length of TSDs is occasionally observed.

Line 159 – “fewer” rather than “less”

Response:

Corrected.

Line 228-233 Figure 4 – colour the name labels with the colour allocated in the legend.

Response:

We updated the figure with colored labels.

Reviewer 2 Report

Here, the manuscript titled as IS481EU shows a new connection between eukaryotic and prokaryotic DNA transposons”.  The bioinformatic methods used in the manuscript is sound, and most results and major conclusion are convincing and interesting. However, some key results were not well organized/presented. Here I have some comments and major concerns, which may improve the manuscript.

Major concerns:

1.      Abstract should clearly show species containing IS481EU, their structure organization (truncated or full copy?) and copy number in each genome for each element.

2.      The structure organization of IS481EU was not reported, how about TIR, transposases, total lengths of these elements and their variations in these species, which is very interesting to readers.

3.      The sequence lengths, variation, and similarity of transposases encoded by IS481EU, their alignment and domain information, and comparison with other relative families (such as ginger) were not reported in results, although the phylogenetic tree was presented. This section also should be presented in abstract.

4.      Introduction missed several closely relative references about evolution of mobile elements between eukaryote and prokaryote (TnpB/IS605, IS607, and Sailor/DD82E)

Author Response

We would like to thank all reviewers and the editor for the thorough review of our manuscript. We have read the comments carefully and following these suggestions, we have revised the manuscript. 

Major concerns:

  1. Abstract should clearly show species containing IS481EU, their structure organization (truncated or full copy?) and copy number in each genome for each element.

Response:

We added the description of IS481EU with some more details. Also, we added a table as a supplementary file and there we include the information on TSDs, TIRs, total length and the length of transposases, the average identity to the consensus sequence, and the copy numbers of insertions having both termini.

  1. The structure organization of IS481EU was not reported, how about TIR, transposases, total lengths of these elements and their variations in these species, which is very interesting to readers.

Response:

Thank you for the suggestions. We added a table as a supplementary file and there we include the information on TSDs, TIRs, total length and the length of transposases, the average identity to the consensus sequence, and the copy numbers of insertions having both termini.

  1. The sequence lengths, variation, and similarity of transposases encoded by IS481EU, their alignment and domain information, and comparison with other relative families (such as ginger) were not reported in results, although the phylogenetic tree was presented. This section also should be presented in abstract.

Response:

We added a table as a supplementary file and there we include the information on TSDs, TIRs, total length and the length of transposases, the average identity to the consensus sequence, and the copy numbers of insertions having both termini. Figure S1 shows the alignment of representative transposase domains from all related transposons.

  1. Introduction missed several closely relative references about evolution of mobile elements between eukaryote and prokaryote (TnpB/IS605, IS607, and Sailor/DD82E)

Response:

Thank you for pointing them out. Because we focus on DNA transposons with a DDD/E transposase, we added the citation of Sailor/DD82E, which is a lineage of the Mariner/Tc1 superfamily. TnpB/Is605 and IS607 do not encode a DDD/E transposase, so we do not introduce these transposons in the manuscript.

Round 2

Reviewer 2 Report

most conerns have been addressed.